# Parents as First Responders: Experiences of Emergency Care in Children with Nemaline Myopathy: A Qualitative Study

**DOI:** 10.3390/nursrep15080271

**Published:** 2025-07-29

**Authors:** Raúl Merchán Arjona, Juan Francisco Velarde-García, Enrique Pacheco del Cerro, Alfonso Meneses Monroy

**Affiliations:** 1Red Cross University School of Nursing, Autonomous University of Madrid, 28003 Madrid, Spain; raul.merchan@cruzroja.es; 2Research Group in Social Health Care Needs for the Population at Risk of Exclusion, Red Cross University School of Nursing, Autonomous University of Madrid, 28003 Madrid, Spain; 3Research Nursing Group of Instituto de Investigación Sanitaria Gregorio Marañón (IiSGM), 28007 Madrid, Spain; 4Research Group of Humanities and Qualitative Research in Health Science of Universidad Rey Juan Carlos (Hum&QRinHS), 28922 Alcorcón, Spain; 5Department of Nursing, Faculty of Nursing, Physiotherapy and Podiatry, Universidad Complutense de Madrid, 28040 Madrid, Spain; quique@ucm.es (E.P.d.C.); ameneses@enf.ucm.es (A.M.M.)

**Keywords:** myopathies nemaline, caregivers, emergency medical services, qualitative research, nursing

## Abstract

**Background**: Nemaline myopathy is a rare congenital neuromuscular disease associated with progressive weakness and frequent respiratory complications. In emergency situations, families often serve as the first and only responders. The aim of this study is to explore how parents in Spain care for children with nemaline myopathy during emergency situations, focusing on the clinical responses performed at home and the organizational challenges encountered when interacting with healthcare systems. **Methods:** A qualitative phenomenological study was conducted with 17 parents from 10 families belonging to the Asociación Yo Nemalínica. Semi-structured interviews were performed via video calls, transcribed verbatim, and analyzed using Giorgi’s descriptive method and ATLAS.ti software (version 24). Methodological rigor was ensured through triangulation, reflexivity, and member validation. **Results:** Four themes were identified. First, families were described as acting under extreme pressure and in isolation during acute home emergencies, often providing cardiopulmonary resuscitation and respiratory support without professional backup. Second, families managed ambiguous signs of deterioration using clinical judgment and home monitoring tools, often preventing fatal outcomes. Third, parents frequently assumed guiding roles in emergency departments due to a lack of clinician familiarity with the disease, leading to delays or errors. Finally, the transition to the Pediatric Intensive Care Unit was marked by emotional distress and rapid decision-making, with families often participating in critical choices about invasive procedures. These findings underscore the complex, multidisciplinary nature of caregiving. **Conclusions:** Parents play an active clinical role during emergencies and episodes of deterioration. Their lived experience should be formally integrated into emergency protocols and the continuity of care strategies to improve safety and outcomes.

## 1. Introduction

Although individually infrequent, rare diseases—defined in the European Union as those affecting fewer than 5 in 10,000 individuals—collectively impact more than 300 million people worldwide [1]. Most have a genetic origin, manifest in childhood, and typically follow a chronic, progressive, and disabling course. As such, their impact extends beyond physical health, significantly affecting the emotional and psychosocial well-being of families [2].

Nemaline myopathy (NM) is one such rare congenital neuromuscular disorder characterized by progressive muscle weakness, generalized hypotonia, respiratory compromise, and feeding impairment. Its clinical presentation is highly heterogeneous, ranging from mild forms compatible with long-term survival to severe neonatal variants associated with early mortality [3]. Diagnosis relies primarily on histological findings, with nemaline rods observed in skeletal muscle fibers via trichrome staining or with electron microscopy being the pathological hallmark [4]. The condition demonstrates substantial genotypic and phenotypic variability, involving mutations in multiple genes such as *ACTA1*, *NEB*, *KLHL40*, and *LMOD3*, among others [5].

In its most severe form, neonates may present profound hypotonia and immediate respiratory failure, requiring ventilatory support from birth. As children grow, chronic respiratory complications are common—such as hypoventilation, apneas, recurrent infections, and dependence on non-invasive ventilation or tracheostomy [6]. These are often exacerbated by dysphagia, increasing the risk of aspiration and pneumonia and thereby compromising clinical stability and quality of life [7]. Cardiac involvement, including cardiomyopathy, has also been reported in some cases, worsening the overall prognosis [8].

Acute, life-threatening events may occur suddenly and outside hospital settings, demanding urgent and precise responses from caregivers. The home frequently becomes the primary site of intervention during emergencies, particularly in cases of airway obstruction or episodes of laryngospasm [9]. Consequently, the home frequently becomes the initial site of emergency care. Prolonged daily caregiving leads many parents to acquire essential clinical competencies, such as respiratory monitoring, the operation of medical devices (e.g., ventilators and oximeters), and the administration of resuscitation techniques [10].

Despite this reality, the scientific literature has largely focused on the biomedical dimensions of these diseases, with limited attention to the lived experiences of families—particularly in emergency contexts. Benedetto et al. emphasize the need to recognize informal caregivers as frontline providers, especially under high clinical-risk conditions [11]. The situation is further compounded by the insufficient preparedness of emergency services to respond effectively to rare and complex diseases, exacerbating the vulnerability of these children and their families [12,13].

Recent international consensus efforts have highlighted the need for emergency protocols specifically tailored to neuromuscular disorders. For example, Bacher et al. have provided detailed recommendations for managing acute complications—such as steroid-related crises—in neuromuscular patients, emphasizing the importance of rapid and coordinated responses [14]. Likewise, the European Neuromuscular Centre (ENMC) has introduced structured “emergency cards” as standardized tools to enhance clinical preparedness in acute care settings [15]. These initiatives are consistent with growing evidence pointing to systemic gaps in the recognition and management of rare neuromuscular conditions within emergency departments [16].

Concurrently, a number of recent studies emphasize the urgent need to implement structured psychosocial interventions aimed at supporting the caregivers of children with neuromuscular diseases, as they often experience high levels of emotional burden, chronic stress, and social isolation. Domaradzki and Walkowiak, for instance, reported a significant prevalence of depressive symptoms and feelings of institutional abandonment among the parents of children with Duchenne and Becker muscular dystrophies, reinforcing the need for ongoing psychological support [17]. Similarly, a multicenter study by Somanadhan et al. in Ireland identified access to peer support groups and personalized emotional counseling as top priorities to strengthen family resilience and coping capacities [18]. In alignment with this, Brødsgaard et al. demonstrated the benefits of including parents in family-centered care models during prolonged hospitalizations, noting that such involvement helps reduce anxiety, promote the development of technical skills, and improve the overall care experience [19]. In addition, the ENMC advocates for a comprehensive support framework that includes psychoeducational programs, access to respite services, and legal and social counseling to improve the quality of life for families and mitigate caregiver burnout [15].

This study aims to explore how parents in Spain care for children with NM during emergency situations, with particular emphasis on the clinical interventions carried out at home and the organizational challenges encountered when engaging with the healthcare system. The central research question guiding this inquiry was the following: How do parents of children with NM experience and manage emergency situations?

## 2. Materials and Methods

### 2.1. Design

A qualitative study was conducted following a descriptive phenomenological approach rooted in the philosophical framework of Edmund Husserl. The objective was to gain a deep understanding of the lived experiences of parents whose children were diagnosed with NM [20,21]. This methodology emphasizes participants’ subjective perspectives, employing phenomenological reduction to bracket researcher assumptions and focus on detailed, context-rich accounts of everyday caregiving realities [22].

### 2.2. Research Team and Reflexivity

The research team comprised four male nursing professionals and university faculty members with prior experience in qualitative research. Although one researcher was personally acquainted with a participant, no prior professional relationship existed. At the outset of the study, two reflexive sessions were conducted to articulate the team’s positionality, during which theoretical frameworks, personal beliefs, and motivations were openly discussed and acknowledged [23].

The study was guided by an interpretive paradigm, aiming to explore the phenomenon through which participants lived, understood as socially constructed knowledge. The team recognized that NM, due to its rarity and clinical complexity, creates substantial vulnerability for families, particularly in life-threatening situations requiring rapid, autonomous parental response in the absence of immediate professional support. Additionally, systemic gaps were identified in home-based care, the continuity of services, and the readiness of emergency professionals.

This research was motivated by the need to deepen our understanding of how parents experience and make sense of caregiving when faced with critical conditions. By amplifying their voices, this study aims to inform more responsive healthcare practices and support systems tailored to the unique challenges of rare pediatric diseases.

### 2.3. Context and Setting

This study was conducted in Spain and focused on families affiliated with the Asociación Yo Nemalínica, a national non-profit organization that provides support to individuals diagnosed with NM and their families [24]. Spain’s healthcare system follows a decentralized model, with health service responsibilities distributed across 17 autonomous regions. This structure results in significant regional disparities in the availability and accessibility of specialized care for rare diseases such as nemaline myopathy [25]. Given the complexity of the condition, effective home-based management requires coordinated, multidisciplinary care involving various medical specialties [26].

At the national level, four hospitals in Spain have been designated as reference centers for neuromuscular diseases: Hospital Sant Joan de Déu (Barcelona), Hospital Universitario 12 de Octubre (Madrid), Hospital Universitario La Fe (Valencia), and Hospital Virgen del Rocío (Seville). These institutions provide advanced diagnostic and therapeutic services, home ventilation programs, and coordinated referral systems. However, geographic and administrative barriers continue to hinder equitable access to these specialized resources across regions [27].

Although PICU (Pediatric Intensive Care Unit) professionals in Spain receive specialized training in critical care, there remains a lack of education and clinical protocols specifically adapted to rare neuromuscular diseases such as NM. This gap frequently results in delayed or suboptimal treatment and increased reliance on parental expertise—an issue that has also been documented internationally in the management of acute neuromuscular conditions in children [28].

Most PICUs in Spain permit 24/7 parental presence and formally promote family-centered care. Nevertheless, the implementation of this is often inconsistent, particularly with regard to caregiver participation during life-threatening events. Practices in such scenarios vary widely depending on institutional policy and individual professional judgment [29].

### 2.4. Sampling Strategies and Participants

The participants were parents of children diagnosed with NM, many of whom had acquired advanced caregiving skills over time, including first aid and cardiopulmonary resuscitation, often through training provided by healthcare professionals, patient associations, or specialized programs. The level of formal training varied among participants; however, several parents reported gaining practical experience and receiving informal guidance throughout their caregiving journey. Notably, none of the participants reported having received instruction involving artificial intelligence tools. This contextual information helps to define the caregivers’ competency and level of preparedness within the scope of this study.

This study included 17 parents (10 mothers and 7 fathers) from 10 families residing in Spain, all of whom had children under the age of 18 diagnosed with NM and were members of the Asociación Yo Nemalínica. Although participation was open to other family members, only parents ultimately took part in the study. Recruitment began after the project was publicly introduced during the 5th National Meeting of the Asociación Yo Nemalínica in November 2021.

A purposive sampling strategy was employed, selecting participants based on their direct caregiving experience rather than clinical representativeness, which is in line with the study’s phenomenological aims [30]. Sampling continued until thematic saturation was reached—defined as the point at which no new themes or relevant insights emerged from additional interviews [31]. Saturation was monitored and assessed by the research team throughout the concurrent process of data collection and preliminary analysis. After each interview, transcripts were reviewed, and emerging themes were systematically compared across participants. The decision that thematic saturation had been reached was made collectively by all four researchers during an analytic meeting following the 17th interview, at which point no substantially new categories were identified. This decision-making process was documented in a shared research log, where evolving themes, coding patterns, and reflexive notes were systematically recorded to track the study’s progression toward saturation. All participants completed the study, and no dropouts were recorded.

### 2.5. Data Collection

Between January 2022 and June 2023, qualitative data were collected through semi-structured interviews incorporating open-ended questions, allowing for an in-depth exploration of participants’ experiences and perspectives (see the full interview guide in Appendix A Table A1).

Each participant completed a single interview session conducted remotely via Microsoft Teams, a platform chosen for its accessibility and alignment with current standards in digital qualitative research [32,33]. Individualized invitations were sent via email, and informed verbal consent was obtained prior to each session. With participants’ approval, both the audio and video were recorded. This dual-recording method enabled the capture of both verbal content and non-verbal cues—such as facial expressions, eye movements, and body posture—providing essential context for emotional and experiential interpretation. In parallel, researchers documented field notes to register behavioral observations, environmental context, and methodological reflections [34].

During data collection, no formal technical consultations or psychological support sessions were conducted as part of the interviews. However, interviewers adopted a supportive and empathetic stance, fostering a safe and respectful environment that encouraged participants to openly share their experiences. Any signs of emotional distress were handled with sensitivity, and participants were reminded of their right to pause or withdraw from the interview at any time. This approach contributed to the richness and depth of the qualitative data obtained.

The interview process generated a total of 1246 min of recorded material, with individual sessions ranging in duration and averaging 74.6 min (SD ± 29.1).

### 2.6. Analysis

A comprehensive analytic strategy was employed, beginning with the verbatim transcription of each interview and the compilation of inductive field notes. To facilitate systematic coding and data organization, the research team utilized ATLAS.ti (version 2024), which is a qualitative analysis software.

Thematic analysis was conducted following Giorgi’s descriptive phenomenological method, which aims to capture the essential structure of participants’ lived experiences while minimizing researcher interpretation [35]. The process unfolded in four stages: First, transcripts were read holistically to grasp the overall narrative and emotional tone; second, a detailed line-by-line analysis was conducted to identify significant units of meaning, which were coded using ATLAS.ti. These units were then clustered into thematic categories based on recurrent patterns and conceptual coherence. Finally, categories were synthesized into overarching themes that reflected the key dimensions of the caregiving experience among the families of children with NM.

Each transcript was initially analyzed independently to preserve the authenticity of individual narratives. Coding was performed collaboratively, and discrepancies were resolved through team consensus to ensure methodological rigor, analytical consistency, and the credibility of the findings.

### 2.7. Rigor Criteria

This study adhered to established guidelines for qualitative research, including the Consolidated Criteria for Reporting Qualitative Research (COREQ) [23], the Standards for Reporting Qualitative Research (SRQR) [36], and Guba and Lincoln’s trustworthiness criteria [37]. A series of methodological strategies were employed to enhance the rigor, credibility, and transparency of the findings. Investigator triangulation was achieved through independent coding and collaborative analysis, while methodological triangulation combined semi-structured interviews with field notes. To ensure the accuracy and credibility of the data, a member-checking process was conducted after the transcription of the interviews. Each participant was invited to review the full transcript of their interview to confirm that their statements had been accurately represented. They were encouraged to suggest corrections or clarifications in the case of any inaccuracies or misinterpretations. Minor wording adjustments were made based on their feedback; however, no substantial modifications were required. This procedure enhanced the authenticity of the data and contributed to the confirmability of the study’s findings.

Transferability was supported by providing rich, contextualized descriptions of participants, settings, and research procedures, enabling readers to assess the applicability of the results to other contexts. Dependability was reinforced through an external audit conducted by an independent researcher. Confirmability was addressed through reflexive practices, including positionality statements and researcher journaling. These integrated strategies contributed to the overall trustworthiness of the study and anchored the findings in the lived experiences of participants.

### 2.8. Ethical Considerations

This study was conducted in accordance with the ethical principles outlined in the Declaration of Helsinki. Participation was entirely voluntary. All participants received detailed information regarding the objectives, procedures, potential risks, and benefits of the study and provided informed verbal consent prior to data collection. They were also informed of their right to withdraw from the study at any point without consequences.

To protect privacy and ensure confidentiality, all interview data were anonymized and stored securely in password-protected files accessible only to the research team. Identifying information was removed from transcripts and replaced with pseudonyms. Audio and video recordings were permanently deleted after transcription and analysis were completed.

Given the emotionally sensitive nature of the interviews, particular care was taken to foster a respectful and supportive environment throughout the data collection process. Researchers were trained in empathetic listening and monitored participants’ emotional well-being during the interviews. When necessary, participants were reminded of their right to pause or discontinue the session at any time.

## 3. Results

The majority of participants were married (64.7%), had one or two children (41.2% each), and resided in urban areas (70.6%) in owner-occupied housing (88.2%). Most households owned a vehicle (70.6%) and reported having university-level education (70.6%). Employment status varied, with 17.6% of participants on parental leave. The monthly household income was highest in the EUR ≥ 2000 bracket (41.2%). The gender distribution among children was evenly split between males and females (Table 1). From the interview data, four major themes emerged (Figure 1). Selected excerpts from participants’ narratives are presented to illustrate and contextualize each theme.

### 3.1. Theme 1: Out-of-Hospital Emergencies: Caring in Isolation and Under Pressure

Home-based care for children with NM places significant demands on families, where parents frequently act as first responders to sudden clinical deteriorations without immediate professional support and with no margin for delay. In these contexts, the home becomes the primary setting for life-sustaining interventions. M1 underscores the profound sense of isolation inherent in at-home care: unlike the hospital environment, there is no support network at hand, and the burden of critical decision-making often falls solely on the parent.


*“…if you’re in the hospital, you try to solve it, and if not, you press a button and the nurse comes running. At home, you can’t press a button—you’re alone with yourself.” *

*(M1)*


This reality is vividly illustrated in P2′s account of responding to his child’s cardiopulmonary arrest at home. He acted swiftly, performing CPR and initiating BiPAP ventilation before the emergency services arrived. His narrative reflects both the technical competence and intense emotional pressure that families endure during high-stakes emergencies.


*“H2 was taking a bath, I was alone with both kids, his sister was playing and pulled his hand, and he fell with his head into the water. I don’t know how I reacted so fast—I called 112, put the phone on speaker; he turned very purple and I couldn’t find a pulse, so I started doing chest compressions. H2 came back, his color improved. I kept talking to them, explaining everything. They sent an ambulance; meanwhile, I placed the BiPAP and his oxygen started going up—by the time they arrived, it was at 88–90.” *

*(P2)*


Furthermore, the critical nature of these episodes frequently extends beyond the initial response and into the hospital transfer process. P5 describes the distressing experience of an ambulance transporting her clinically unstable daughter suddenly stopping on the highway, creating a moment of extreme uncertainty and fear.


*“…she became very unstable at the hospital, and the transfer could be risky—even they were uncertain. We were driving ahead in our car, and the ambulance was following behind. At one point, the ambulance just stopped in the middle of the highway. It was very dramatic.” *

*(P5)*


Similarly, M8 highlights a systemic weakness: the lack of preparedness among some emergency professionals to manage complex cases of NM. Following severe decompensation and inadequate initial care, her daughter required urgent admission to a PICU for stabilization, an outcome she describes as “a miracle.”


*“…I called the ambulance in a panic… we were so nervous, so scared, I told the person on the phone, please, my daughter is dying. A doctor came who had no idea what to do, gave her oxygen, and we transferred her to the PICU—they told us she had been saved by a miracle.” *

*(M8)*


Collectively, these narratives depict the family as the cornerstone of emergency responses, often acting in isolation and without structured support. In the absence of timely medical intervention and professional readiness, it is the parents’ intuition, technical skills, and unwavering commitment that become vital to the child’s survival.

### 3.2. Theme 2: Decision-Making Under Pressure: Managing Clinical Deterioration at Home

Beyond the urgent interventions already described, many of the families of children with NM face an additional layer of complexity: the need to identify, interpret, and manage ambiguous clinical signs that—if not addressed in time—can escalate into life-threatening situations. These events often do not begin as obvious crises but rather as subtle signs of decline that require constant attention, clinical judgment, and decisive action in highly uncertain conditions.

M4′s experience is illustrative in this regard. She refused her daughter’s hospital discharge, guided by her intuition that something was not right. Shortly thereafter, the child suffered a cardiac arrest, confirming the seriousness of the condition that had been underestimated by the medical team. Her determination was crucial in saving her daughter’s life. A common pattern across multiple accounts is the silent progression of respiratory deterioration, which can go unnoticed even by healthcare professionals.


*“If I hadn’t been so stubborn, my daughter wouldn’t be here. They wanted to send her home, but I kept saying something wasn’t right—I refused to leave. Soon after, her heart rate started climbing, her oxygen dropped, and she went into cardiac arrest. They had to resuscitate her. That image is burned into my memory, no matter how much I want to forget it.” *

*(M4)*


This was not an isolated incident. On another occasion, M4 recalls how her daughter began to show drowsiness and cyanosis while at school. Upon admission to the PICU, she was found to have an oxygen saturation of 65% and a CO_2_ retention of 210, revealing the severity that may lie behind subtle clinical signs.


*“One day I took her to school, and she fell asleep. When we realized it, her fingernails were blue—she was running out of oxygen. She had been retaining CO_2_ for nearly a month. That was her first hospital admission. At two years old, her oxygen dropped to 65 and her CO_2_ went up to 210. She was asleep for four days in the PICU.” *

*(M4)*


In other situations, continuous home monitoring helped identify imminent risks. M8, for example, observed nonspecific signs such as fatigue, distress, and loss of appetite. Using a pulse oximeter, she discovered a dangerously low oxygen saturation of 68%. These episodes highlight not only the importance of vigilant home monitoring but also parents’ ability to anticipate clinical deterioration—often without direct clinical support. The home use of medical technologies and devices also involves significant risks.


*“At home, my daughter said she felt anxious, unwell, hadn’t wanted to eat for days, just wanted to lie down—she was very tired. I checked her with the pulse oximeter, and she was at 68.”*


The home use of medical technologies and devices also carries significant risks. This is exemplified by P3 and M7, who describe critical incidents caused by ventilator failures or errors in nasogastric tube feeding, demonstrating that even experienced caregivers must contend with system fragility and ongoing uncertainty.


*“H3 had a crisis, and the ventilator disconnected. It failed, and by the time we noticed, he was already desaturating.” *

*(P3)*



*“Many times I would insert the feeding tube through his nose, and I didn’t know if it was reaching his stomach… He choked twice—one of them was really serious. I was alone and thought he wasn’t going to make it… he started aspirating until it turned into aspiration pneumonia.” *

*(M7)*


Finally, accounts such as M2′s reveal the lasting emotional impact of these situations. After performing CPR on her child at home, she described it as the worst moment of her life, emphasizing that only her ability to act quickly prevented a tragedy. Her testimony reflects the immense responsibility that families must shoulder in such moments.


*“I’ve never experienced anything worse in my life. I was doing chest compressions, and he was turning grey. I kept thinking, ‘Oh my God, this can’t be happening to me, please.’ I completely broke down when the ambulance arrived—I cried all the way to the hospital. If another parent didn’t know how to react in that moment, their child wouldn’t survive.” *

*(M2)*


Far from being exceptional, these experiences represent the daily reality for many families who, in the absence of immediate medical support, develop essential clinical and technical skills. In this context, continuous vigilance and early intervention become vital forms of caregiving.

### 3.3. Theme 3: Shared Emergencies: When Families Guide Professional Care

Pediatric emergency care plays a central role in the healthcare journey of families with children diagnosed with NM. Although emergency services are designed to respond rapidly to critical situations, the testimonies collected reveal that they are not always prepared to address the complexity of rare conditions. In response to this gap, parents often cease to be mere companions and instead become clinical interpreters, mediators between lay knowledge and professional expertise, and active monitors of potential system failures.

M1 illustrates how emergency visits became routine to the point that she knew exactly what to say to ensure her child received the appropriate care. However, such expertise is not always acknowledged.


*“We go to the emergency room a lot, and we know what needs to be said—you know when things are going wrong.” *

*(M1)*


M8 recounts the distress of being assisted by a professional unfamiliar with NM, which led to a critical delay in care.


*“They sent a pediatrician from the health center who didn’t know anything about NM and told me there was nothing he could do—it devastated me. He was unable to help.” *

*(M8)*



*“They told us it was just gastroenteritis… but it was actually a bowel obstruction. By the time they realized, it was too late… she ended up in emergency surgery.” *

*(M8)*


Similarly, M9 describes how her concerns about her child’s respiratory symptoms were dismissed until an objective test finally prompted medical intervention. These accounts reveal a systemic failure to engage in active listening, often minimizing the experiential knowledge that families develop through daily caregiving.


*“I kept telling them my child’s oxygen was low, and the doctor said he was fine. Then they used the pulse oximeter, and it read 88 and dropping… that’s when they realized he wasn’t doing as well as they thought.” *

*(M9)*


In some cases, such as M7′s, hasty clinical decisions made under pressure—like premature hospital discharge—ultimately shift the burden of managing complications back onto the family.


*“…we went into the ER in very, very bad shape. The ER was overwhelmed, and they discharged us. The trip home was awful—it was so hard because I could see my son’s lips changing color, but I had no idea what was happening… something told me that color wasn’t normal…” *

*(M7)*


Collectively, these narratives highlight that caregivers take on an active, technically informed, and emotionally demanding role within a system that frequently fails to recognize them as integral to the care process. In the face of institutional unpreparedness to manage rare diseases like NM, it is the families—through their lived experience, commitment, and capacity to act—who play a significant role in emergency care delivery.

### 3.4. Theme 4: From Emergency to Intensive Care: Challenges in Ensuring the Continuity of Care

Emergency care for children with NM is not an isolated event but often marks the beginning of a complex clinical journey. Admissions to the PICU are frequently an inevitable extension of these acute episodes, during which families assume the role of expert observers, and medical decisions must be made quickly. P3 describes how emergency visits became a recurring prelude to PICU admissions, eventually forming part of a routine in which they have learned to navigate hospital procedures, equipment, and dynamics.


*“Being readmitted felt like coming home—you’d arrive at the PICU, and they treated you like one of their own. After so much time there, you know how everything works.” *

*(P3)*


Respiratory complications are one of the main reasons for admission. M1 recounts how, following a cannula change, her daughter developed persistent tachycardia, prompting an emergency visit. There, she was diagnosed with a collapsed lung secondary to bronchomalacia, leading to immediate PICU admission.


*“After a cannula change, she came home with severe tachycardia. I called the complex chronic care unit, and they told us to come back. We went through the ER, and from there straight to the PICU… her left lung had collapsed… it was bronchomalacia in the upper left bronchus.” *

*(M1)*


Similarly, P4 reports several episodes of respiratory failure and ventilator malfunction that also led to intensive care intervention.


*“Her more severe symptoms started around age two… she had several PICU admissions up to age three—three or four very serious episodes of respiratory failure, and also problems with the ventilator.” *

*(P4)*


Even when these admissions become familiar, their emotional impact does not diminish. M8′s testimony reveals the life-threatening fragility of these children; during a PICU stay, she witnessed her daughter “go lifeless,” in her words. The medical team’s rapid intervention was essential to reverse the situation, but the fear and vulnerability of that moment left a lasting mark.


*“She had a PICU admission where she just went lifeless… Thank goodness they were there, they reacted immediately, and later the nurses told us how bad it really was.” *

*(M8)*


In a similar vein, M7 describes not only the urgency of her child’s respiratory decompensation but also the ongoing emotional toll of the process. She characterizes the experience as “very, very traumatic” and details the subsequent medical decisions they faced: PEG placement, a failed extubation attempt, and ultimately, a tracheostomy. Her story illustrates how a single emergency can trigger a cascade of interventions that profoundly reshape family life.


*“A PICU admission is very, very, very traumatic—until everything stabilizes… that’s when they scheduled the PEG. There was a failed extubation attempt, and in the end, of course, a tracheostomy and invasive ventilation.” *

*(M7)*


## 4. Discussion

The findings of our study reveal that the home becomes the primary, and often only, setting for emergency medical care for children with NM. In this context, parents—especially mothers—are the first to attend to their children in urgent situations without the support of healthcare professionals. This situation aligns with the results of Satchell et al., who found that families act as first responders and coordinators in emergency situations, assuming a high emotional and caregiving burden [38]. In this regard, the role of the informal caregiver not only requires specialized technical skills but also has a significant impact on the mental health of those performing it. Along these lines, Yao et al. report that more than 70% of informal caregivers experience a high level of burden, accompanied by symptoms of anxiety, depression, and chronic stress, which directly affects their physical and emotional well-being [39]. These data underscore the urgent need to implement structured psychological support interventions specifically aimed at family caregivers facing highly complex care situations in the home setting.

Moreover, parents described inadequate responses from emergency services, attributed to the lack of specific protocols and limited training of personnel in neuromuscular diseases. In this regard, Racca et al. propose the implementation of emergency cards containing essential clinical information about the condition and the patient’s individual needs as a corrective measure, thus facilitating faster, more effective, and personalized care [40]. In the absence of immediate professional support, parents developed advanced skills to identify subtle clinical signs and make critical decisions in the daily management of the disease.

Our findings are consistent with previous research on caregiving in rare neuromuscular and musculoskeletal disorders, such as Duchenne muscular dystrophy, Pompe disease, arthrogryposis multiplex congenita, and other progressive neuromuscular conditions. Across these contexts, parents—particularly mothers—are often compelled to assume roles that extend beyond emotional caregiving, performing highly technical tasks such as managing ventilators, administering medications, and identifying early signs of decompensation [11,41,42]. These responsibilities are frequently taken on in the absence of adequate healthcare support, which increases caregiver stress and fosters a sense of isolation and clinical over-responsibility [43]. In all cases, the lack of condition-specific protocols and the variability of professional preparedness contribute to parents acting as first responders and care coordinators. Compared to better-known rare diseases, families of children with NM often face greater diagnostic uncertainty, limited system awareness, and fragmented care pathways, placing them in a particularly vulnerable position during emergencies.

As highlighted by Baker and Claridge, mothers take on an active role that turns them into true experts in their children’s medical condition, acting as primary advocates in both clinical and educational settings and facing the many challenges posed by caring for a complex chronic illness with resilience [44].

In turn, the use of medical technology in our study indicated the improved care of children within the home but also the transfer of complex clinical responsibilities to families, requiring ongoing training, the capacity for emergency response, and appropriate technical support. The existing literature consistently highlights that many parents acquire advanced competencies in managing devices such as ventilators and tracheostomies [18,45], often facing high levels of stress in critical situations without sufficient professional support [46,47]. These conditions emphasize the need to establish emergency protocols and specialized support networks [48,49]. Another particularly relevant finding is the constant vigilance of parents, who developed the ability to identify subtle clinical signs and anticipate decompensation. This home-based monitoring enabled the early detection of significant changes, contributing to the prevention of hospitalizations [50]. However, constant caregiving by parents can result in a high emotional burden, manifested as anxiety, fatigue, and isolation. Recent studies have documented this impact; for example, Toledano-Toledano and Luna identified caregivers with high levels of stress, anxiety, and limited family support during hospitalizations [51], while Cormican and Dowling emphasized how the caregiving role transforms daily life and generates profound emotional isolation [52]. These findings underscore the urgent need to address the psychosocial needs of caregiving families.

Our study showed that visits to pediatric emergency departments generate tension and vulnerability for parents due to the lack of preparedness of healthcare professionals in dealing with NM, forcing them to act as intermediaries between the physician and the child. Pelentsov et al. warn that caregivers of children with rare diseases acquire valuable clinical knowledge through their daily experiences, but this expertise is often disregarded in hospital settings, potentially leading to clinical errors and delays in care [53]. Verberne et al. note that the exclusion of caregivers’ knowledge in critical situations increases anxiety and a sense of clinical helplessness [54]. Boivin et al. emphasize the importance of recognizing caregivers as “co-producers” of clinical knowledge, given their key role in the early identification of signs that protocols may overlook [55]. Carel & Kidd describe this phenomenon as a situation in which the caregiver’s voice is systematically devalued, revealing an imbalance in care where clinical responsibilities are delegated without proper support or recognition [56]. Furthermore, the lack of specific training among emergency staff regarding rare diseases such as NM can lead to diagnostic delays, treatment errors, and an overreliance on parental judgment, all of which negatively impact caregivers’ mental health [57]. The absence of specific protocols for neuromuscular diseases in emergency services remains one of the main barriers to adequate care for these patients [58].

Finally, the transition from the emergency department to the PICU represents a critical moment in the clinical management of children with NM, given the frequent onset of various health complications. For many families, hospital admissions become a recurrent and familiar experience. Admission to the PICU not only involves increasing clinical complexity but also significantly heightens parental distress [59]. In this setting, families must quickly assume an active role in care, acquiring essential technical skills for managing medical devices. This learning process, while necessary, unfolds in an atmosphere of high tension and anxiety. According to Brødsgaard et al. [19] and Lean et al. [59], training in complex care during hospitalization is essential to ensure a safe transition home, turning the PICU into a space not only of high emotional burden but also of intensive learning for families. This educational process becomes even more relevant during prolonged hospitalizations, where caregivers must rapidly acquire technical competencies within a limited timeframe [60]. In parallel, the literature supports the active involvement of parents in the PICU, positioning it within the Family-Centered Care (FCC) model. This approach promotes the inclusion of caregivers in clinical decision-making and care delivery, recognizing their experiential knowledge and crucial role in ensuring the safety of the hospitalized child. According to Butler et al., such collaboration improves the quality of care, enhances the early detection of complications, and helps mitigate the emotional impact of hospitalization for both children and their families [61]. Ultimately, admission to the PICU is experienced as a traumatic event, marked by urgency, fear, and significant clinical decisions, such as the placement of a PEG or a tracheostomy, that profoundly disrupt daily life. This experience reinforces the fact that the PICU represents not only an increase in clinical complexity but also an intense emotional transition in which collaboration between healthcare professionals and families is essential to sustain care and humanize the healthcare process.

This study has several limitations that should be considered. As a qualitative investigation with a small sample focused on a specific healthcare context (Spain), the results are not generalizable; however, they do offer a deep understanding of family experiences. The predominance of mothers as participants may limit the diversity of perspectives and the use of interviews may have been influenced by emotional or relational factors. Moreover, recruiting participants exclusively through a patient association may have introduced a self-selection bias, as families with greater involvement, awareness, or access to support networks could be overrepresented. This might have limited the inclusion of more diverse caregiving experiences, particularly from families who are not connected to organized associations or advocacy groups. Likewise, the gender imbalance in the sample may reflect caregiving patterns but also shape the type of narratives collected. These factors should be considered when interpreting the scope and transferability of the findings. Additionally, the absence of healthcare professionals’ perspectives prevents a more comprehensive understanding of the clinical management of NM.

Beyond these limitations, this study contributes conceptually to ongoing discussions in healthcare theory and offers valuable insights for future methodological approaches. First, our findings support the idea of “distributed care” [62], where responsibility for managing complex conditions like NM extends from clinical institutions into the home, with families assuming critical tasks. The phenomenon of the “medicalization of the home” is evident as parents, particularly mothers, assume tasks traditionally performed by healthcare professionals—such as managing ventilatory support and responding to emergencies. Mah et al. documented how families caring for children with neuromuscular disorders and home ventilation become their child’s “lifeline,” performing complex medical care in domestic settings [46]. This shift highlights how the home increasingly functions as a clinical space, requiring both emotional and technical capacities from caregivers.

Second, this study illustrates how families develop what Epstein termed lay expertise, transforming lived experience into actionable clinical knowledge [63]. Yet, this knowledge often remains invisible or undervalued within institutional settings. Drawing on the concept of epistemic injustice [56] and reinforced by recent contributions [64,65], our findings show how caregivers’ experiential insights are frequently dismissed by professionals, leading to moral distress, diagnostic delays, and avoidable errors.

Third, although FCC remains a foundational principle in pediatrics, our findings indicate that its application in emergency settings is frequently inconsistent. When caregivers are treated merely as passive informants rather than as active partners in care, the transformative potential of FCC is diminished. Jonas et al., in their critical scoping review, highlight how communication barriers and the dismissal of caregiver knowledge contribute to epistemic injustice in clinical encounters. Their work underscores the importance of recognizing the relational and epistemic dimensions of clinician–caregiver interactions as a prerequisite for equitable, collaborative care [66].

These theoretical perspectives help situate our empirical findings within broader debates on healthcare ethics, power asymmetries, and participatory models of care. These studies call for a deeper transformation of clinical cultures—moving from merely acknowledging the caregiver’s presence to truly integrating their knowledge and roles into healthcare processes.

From a methodological standpoint, future research should build on these contributions through alternative and complementary approaches. While this study employed a qualitative design to explore lived experiences in depth, mixed-methods research could help quantify caregiver burdens, stress, and resilience while retaining narrative complexity. Ethnographic or observational studies may provide richer insights into how families manage emergencies in real time, particularly in home settings. Longitudinal designs could also trace the evolution of caregiving roles, needs, and emotional impacts over time. Furthermore, incorporating the perspectives of healthcare professionals alongside those of families could enrich our understanding of care dynamics and help address system-level gaps.

Furthermore, there is currently no dedicated international registry specifically for NM, and no widely established, disease-specific emergency management guidelines are available. Families often receive individualized training and education upon discharge, supported by multidisciplinary teams and patient associations, but standardized protocols and educational tools remain lacking. This gap underscores an urgent need to develop registries and clear management guidelines to enhance caregiver preparedness and patient safety.

One of the main implications for clinical practice that has emerged from our study is the urgent need to establish specific emergency protocols for neuromuscular diseases, as well as to improve the training of healthcare personnel in emergency and PICU settings, valuing caregivers’ experiential knowledge within the care process. Similarly, the importance of implementing comprehensive programs that combine technical training, emotional support, and ongoing assistance networks is emphasized, positioning both the home and the PICU as key settings for care and learning.

In addition, we propose concrete, practical implications based on the findings of this study. First, we consider it essential to develop a standardized emergency card for children with neuromuscular diseases, including key clinical information—such as diagnosis, devices in use, allergies, and recommended action protocols—to support healthcare professionals in responding quickly and appropriately during critical situations. Second, we suggest establishing structured hospital transfer protocols that account for coordination between prehospital and hospital teams, as well as the specific needs of the patient and family. These tools could significantly improve safety, reduce delays, and help humanize care in high-complexity settings.

In line with these implications, several lines of research are proposed to enhance clinical practice and inform public policy design. Future research should focus on evaluating psychosocial interventions for informal caregivers, assessing clinical tools such as emergency cards, and studying parental training during prolonged periods of hospitalizations. Additionally, further exploration is warranted to integrate caregivers’ experiential knowledge into care and decision-making processes. Finally, multicenter research is recommended to examine variations in NM management across autonomous communities, with the goal of identifying best practices and reducing disparities in access to specialized care.

Based on our findings and the literature reviewed, we developed a strategic agenda for clinical and research advancement in this field, which is presented in Appendix B.

## 5. Conclusions

This study underscores the central role that parents play in emergency care and the home-based management of children diagnosed with NM. In the absence of structured healthcare protocols and immediate professional support, caregivers assume the role of first responders, clinical decision-makers, and long-term providers of complex medical care. Their ability to detect subtle signs of clinical deterioration, respond to life-threatening situations, and manage essential medical equipment demonstrates a level of clinical expertise that remains largely unacknowledged by the healthcare system. Families face significant emotional, physical, and logistical burdens, intensified by structural deficiencies such as the insufficient training of healthcare professionals in rare diseases, diagnostic delays, and unequal access to specialized services. Despite these challenges, parents exhibit remarkable resilience and unwavering dedication, developing advanced caregiving skills under high-stress conditions.

## Figures and Tables

**Figure 1 nursrep-15-00271-f001:**
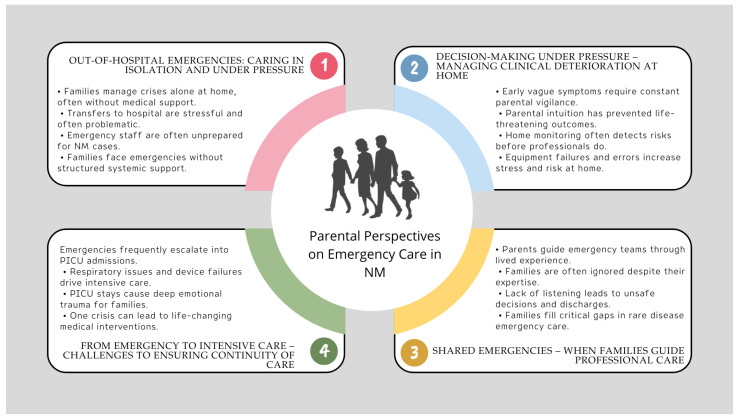
Key themes in parental experiences of emergency care for children with NM.

**Table 1 nursrep-15-00271-t001:** The sociodemographic characteristics of the parents.

Marital Status	Married 64.7%	Single 29.4%	Divorced 5.9%	
Number of Children	1 child: 41.2%	2 children: 41.2%	3 children: 17.6%	
Child’s Sex	50% Female	50% Male		
Type of Residence	70.6% Urban	17.6% Rural	11.8% Semi-urban	
Housing Tenure	88.2% Owned	11.8% Social housing		
Vehicle Ownership	70.6% Yes	29.4% No		
Educational Attainment	70.6% University	17.6% Vocational training	11.8% Secondary	
Employment Status	5.9% Homemaker	5.9% Unemployed	17.6% Parental leave	
Monthly Household Income	41.2% EUR ≥2000	29.4% EUR 1500–2000	17.6% EUR 1000–1500	11.8% EUR ≤1000

## Data Availability

The data that support the findings of this study are not publicly available due to confidentiality agreements and ethical restrictions involving identifiable personal information. However, the data may be made available by the authors upon reasonable request, provided that appropriate measures are taken to ensure the anonymity and privacy of the participating families.

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
