# Peer review of "Parents as First Responders: Experiences of Emergency Care in Children with Nemaline Myopathy: A Qualitative Study"

_nursrep, 2025, doi:10.3390/nursrep15080271_

Round 1

Reviewer 1 Report

Comments and Suggestions for Authors

Dear Authors,

Congratulations on your manuscript titled “Parents as First Responders: Experiences of Emergency Care in Children with Nemaline Myopathy.” Your study addresses a highly relevant and underexplored topic, offering meaningful contributions to pediatric nursing and rare disease care.

To further strengthen your manuscript, I respectfully offer the following suggestions:

  1. Introduction

    • Consider including recent international references concerning emergency protocols for neuromuscular diseases and psychosocial interventions for caregivers.

  2. Methods

    • Please provide more detail on the saturation process (who determined it, how it was documented).

    • Include the full interview guide as a supplementary appendix. Also, indicate whether member checking was performed to validate transcripts.

  3. Results

    • Improve the resolution and explanatory caption of Figure 1. Consider using a color scheme with appropriate contrast to enhance clarity.

    • Standardize the use of decimals in all percentage values (choose between comma or period and apply consistently across all tables).

    • Reformat Tables 3–6 into formal tables with clear row and column structures for easier reading.

  4. Discussion

    • Expand on limitations, particularly regarding the potential for self-selection bias, given all participants were affiliated with a patient association, and the predominance of female caregivers.

    • Include more specific practical implications, such as proposing an emergency card template or a hospital transfer protocol.

  5. Conclusion

    • Suggest future research directions, such as longitudinal studies investigating the long-term psychological impact on caregivers.

Thank you for your valuable contribution. I hope these suggestions are helpful in refining your work.

Author Response

The response letter is attached as a file.

Reviewer 2 Report

Comments and Suggestions for Authors

Dear Authors,

I commend you for undertaking this topic.

To clarify the study background, please consider responding to the following questions:

Could you please specify whether the study subjects had any prior first aid/ CPR training or AI guidance?

Could you please specify whether any technical discussions or psychological debriefings took place with the subjects?

Could you please specify if a registry exists for this specific myopathy, and if there are specific management guidelines for life-threatening events to be taught to parents upon discharge?

Could you please specify whether any medical facilities are dedicated to patients with this myopathy, what level of critical care training the staff possesses, and whether parents are permitted to stay with their children in the hospital/ICU, including whether there are policies allowing parental presence during life-threatening events?

Author Response

(The authors gave the same response as above.)

Reviewer 3 Report

Comments and Suggestions for Authors

Dear Authors,

thanks a lot for the possibility to read your Manuscript.

I think that from the beginning of the paper, considering also the title, is really clear the objective of the study and the scientific soundness of the topic, from a clinical as well as from a social and organisational perspective.

However, I would provide some suggestions in the following to improve the work.

  • The Introduction is well-conceived, with a structure that helps in defining the context of reference, the needs as well as the paper’s objective. Only one little suggestion is to explicitly define and clarify the research question(s) as well as the research hypotheses.
  • In addition, I suggest to revising the first paragraphs of the Methodology not presenting in this section objectives (line 104-107) but perhaps moving these considerations at the end of the Introduction.

Considering Table 1. Question guide181 that reports the structure of the interview in terms of provided questions, it is not clear if the questions are open questions or in some cases multiple options are available. I suggest to better specify this point.

  • The Results’ section is clear defined arounding the main themes that emeged from the interviews. However, reading this section, I’m wondering about the possibility to provide some graphical representations or synthesis to help an external reader and to not create dispersion among the different themes and related quotes.

In addition, I do not recognise any theme specifically dedicated to the “emotional” aspects, even if in the paper’s objective this domain emerges as an important topic (ref. line 79 in the Introduction section). Please clarify the motivation or remove this aspect.

  • Within the Discussion section, I suggest to the authors to compare the achieved results in the light of other similar studies, as well as to discuss if the topic investigated in the study could potentially be assessed in future research with different methodological approaches. This can enrich the paper, focusing not only on the results and on the implications of the study, but also providing theoretical contributions and methodological insights.

I would also suggest to the authors to develop an “Agenda” basing on the achieved results and integrating them with the multiple bibliographic references reported in the Discussion, to allow the definition of future steps, needed in the research as well as in the practice, to support these patients and their relatives.

In the Discussion section potential theoretical implications are not shown.

Good luck for the publication!

Author Response

(The authors gave the same response as above.)

Round 2

Reviewer 3 Report

Comments and Suggestions for Authors

Dear Authors,

I really appreciate the effort dedicated to revise the paper.

All the section are now very clear and well structured. I really like the presence of the two Appendices to show the interview guide and the agenda defined for future publications and works.

There are some grammatical errors in the paper and I suggest a English revision and editing to be sure to adhere to the standard of a scientific journal.

Good luck for the publication!

Author Response

Dear Reviewer,

First and foremost, we would like to sincerely thank you for your kind words and for your positive assessment of the revised manuscript. We truly appreciate your recognition of the improved clarity and structure, as well as your appreciation for the inclusion of the two appendices.

Regarding your suggestion to revise the manuscript for grammatical accuracy, we would like to inform you that we have requested a thorough English language editing from the journal’s official translation and editing service. We have attached the corresponding certificate to confirm that the manuscript has undergone professional language revision.

We are grateful for your thoughtful observation, which has helped ensure that our work meets the high standards of scientific writing required by an international journal.

With kind regards and our sincere thanks for your time and dedication in reviewing our manuscript.

Sincerely,
